# Design of a High-Gap Plant Protection Machine (HGPM) with Stepless Variable Speed and Power Adjustable Function

Zengbin Cai [1,2,†], Dongbo Xie [1,2,†] , Tao Liu [1,2], Peiyu Hu [1,2], Hongji Liu [1,2] and Quan Zheng [1,2,*]

1 College of Engineering, Anhui Agricultural University, Hefei 230036, China
2 Anhui Intelligent Agricultural Machinery Equipment Engineering Laboratory, Hefei 230036, China
* Correspondence: zhengquan@ahau.edu.cn
† These authors contributed equally to this work.

**Abstract:** The passing performance and driving stability performance of HGPM in an unstructured environment will directly affect the quality of HGPM operation. This paper designs an HGPM power chassis transmission system in order to address the problem of poor adaptability of existing plant protection machines to complex working conditions in the field, especially in the middle and late stage of plant protection operation of tall crops, which cannot pass smoothly due to the obstruction of vision and special road conditions resulting in insufficient traction of the whole machine. The system is theoretically analyzed based on hydrostatic transmission and a time-sharing four-wheel drive splitter; then, based on Solidworks and RecurDyn software, the HGPM is modeled in three dimensions, and the dynamic simulation of working conditions such as climbing, crossing the ridge, and opening the road during field operation is carried out. The simulation results show that the 2H mode can climb over a slope with an angle in the range of $0-25°$ and a ridge with height in the range of $0-100$ mm, the 4H mode can climb over a slope with an angle in the range of $0-35°$ and a ridge with height in the range of $0-320$ mm, with relatively stable body speed and the wheel rotation angular speed converging faster under the open road condition. Finally, prototype performance tests were conducted. The test results show that 4H mode can smoothly pass the ridge with a ridge height of 320 mm and a slope of $26°$, while 2H mode has a sharp drop in speed to 0 after a short fluctuation. 4H mode achieves a more rapid convergence of longitudinal wheel stability compared to 2H mode. The developed chassis drive system of a new type of HGPM meets the design requirements and provides a reference for the dynamic chassis design of HGPM.

**Keywords:** HGPM; dynamic chassis; step-less speed; RecurDyn simulation; passing performance



## 1. Introduction

Plant protection machinery is an effective pest control tool that effectively ensures world agriculture sustainability [1,2]. The development of plant protection machinery has experienced three types of manual backpacking, traction to self-propelled. The HGPM, a kind of self-propelled plant protection machinery, is the most widely used. As the working environment is in the process of crop growth and the harshness of the field environment has an enormous impact on its performance, the design of the powertrain is critical [3,4]. Many scholars at home and abroad have conducted a series of studies, from early sprayers and bar sprayers to the present HGPM, and many research results have been obtained [5–7]. In recent years, many scholars have conducted a lot of research on the power transmission system of plant protection machinery. Zhang C et al. [8] proposed a design scheme for the hydraulic chassis drive system for a large plant protection machine. Chen Y et al. designed an "X" type hydrostatic wheel drive system consisting of a closed swashplate axial piston variable pump, a piston motor, and a reduction device [9]. Chen Z et al. [10] studied the travel performance of wheels when driving in complex unstructured soils and proposed a new rigid wheel for agricultural machinery suitable for muddy water fields. Hugo R. et al. [11] adopted zero

moment point and fuzzy logic for agricultural machinery travel control systems. Various drive modes of plant protection machinery have emerged with the rapid popularity of plant protection machinery in agriculture. Li Z et al. [12,13] designed a crawler motor-driven plant protection machinery and analyzed its performance. However, for HGPM, the two most common drive modes are 2WD and four-wheel drive 4WD. The main advantage of the 2WD is lower fuel consumption in the driving process, and its disadvantage is the problem of insufficient power in uphill or over-can conditions [14]. 4WD is mainly divided into two kinds: one is the full hydraulic drive, including hydraulic steering and hydraulic drive, its main function being the management of the parameters of the hydraulic motor to control the performance of the system; the other is a purely mechanical full-time 4WD system. Although the driving force is strong for full-time driving, there is a problem of higher fuel consumption. The on-demand 4WD system will automatically switch to 2WD or 4WD mode according to the vehicle's driving conditions, without human operation. However, the extended application of the on-demand 4WD can lead to overheating and failure of the multi-plate clutch in the timing splitter due to prolonged friction [15,16]. Compared with the previous two 4WD technologies, time-sharing 4WD technology is mature, has a simple structure, good reliability, and does not require the assembly of the central differential lock. The budget is much lower than that required for full-time 4WD because it is more applicable to the power transmission system of the upland gap planters to ensure that they have higher fuel economy and better passability.

Several drive chassis widely used in HGPM, such as fully hydraulic drive systems with a complex system of electro-mechanical–hydraulic integration, are characterized by perfect design, easy operation, and high chassis drive power [17]. CASE IH (Wisconsin, America) Patriot 3230 HGPM is equipped with full-time hydrostatic four-wheel drive. Hagie Manufacturing (Clarion, America) STS16 HGPM is fitted with anti-skid control valves that can increase torque by 15%. John Deere R4030 series self-propelled sprayers are equipped with mechatronic technology that can increase horsepower power to ensure normal driving of planters in complex road conditions [18]. In addition, research on the chassis of sprayers mainly focuses on the drive, steering, suspension system, and frame [19–22]. In line with the above research progress, the power transmission system of HGPM is mainly applied to some agricultural machinery in dry land, forest land, orchard or large farm, domestically and abroad. The relevant research mainly focuses on the full hydraulic drive, steering, suspension system and frame. Little research has reported on the step-less variable speed and adjustable power function of the combination of hydrostatic transmission and time-sharing four-wheel drive system for upland gap planters. In addition, due to the relatively small size of the land in the Yellow Huaihai Plain, it is not suitable for the development of large power chassis of upland gap planters.

The following contributions are reported in this research work:

- This paper designs a chassis power transmission system of HGPM based on hydrostatic transmission and time-sharing 4WD transfer gear and analyzes the walking performance and stability of HGPM in detail, which provides a basis for the development of a chassis power transmission system of HGPM.
- Based on the simulation software, an undercarriage dynamics model of upland gap planters is established for multi-condition performance simulation tests. Then, the real vehicle test is carried out to verify the rationality and realism of the theoretical design of the plant protection machine and related dynamics analysis and other data.
- This paper presents an important reference for engineering applications such as optimizing agricultural machinery dynamics, driving stability, and ensuring fuel economy.

The overall structure of this paper is organized as follows: In Section 2, an HGPM undercarriage power transfer system is designed based on hydrostatic variable speed and time-sharing splitter. In Section 3, computer-based technology is used to carry out simulation tests of the designed prototype for multiple operating conditions. In Section 4, performance tests are conducted to verify the machine passability and travel stability of the HGPM. In Section 5, the power control and the effect of anti-skid braking on operational performance and their effects are discussed. Section 6 presents a summary of the study.

## 2. Dynamics Switching Principle

### 2.1. Whole Machine Structure and Working Principle

In view of the soybean, corn, and wheat cropping patterns and agronomic requirements of the Huang-Huai-Hai Plain, as well as considering the machine field operation efficiency, field walking through, and other factors, in this paper, the chassis structure of the upland gap planters with a power switching function is designed as shown in Figure 1. It mainly consists of a frame, diesel engine, hydrostatic transmission (HST), time-sharing splitter, reducer, steering gear, axle assembly, wheels, medicine box, and dual-channel spraying system. The operator can push and pull the hydraulic shift lever to achieve basic driving functions such as forward, reverse, and infinitely variable speed. In the field operation, the driving force of the HGPM is provided by the diesel engine. Its power transmission route is as follows: First, the engine is transmitted to the HST. Then, the power is transmitted from the drive shaft to the differential through the distributor, and finally, the power is transmitted to the front and rear wheels through the drive axle to realize the self-propelled operation of the HGPM.

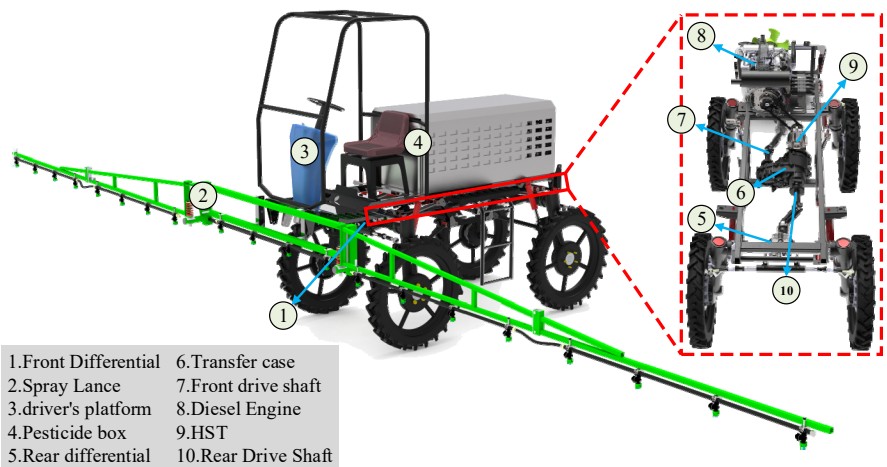

| 1.Front Differential | 6.Transfer case |
|---|---|
| 2.Spray Lance | 7.Front drive shaft |
| 3.driver's platform | 8.Diesel Engine |
| 4.Pesticide box | 9.HST |
| 5.Rear differential | 10.Rear Drive Shaft |

**Figure 1.** The whole machine structure schematic.

### 2.2. Transmission System Design

The performance of the HGPM power chassis depends largely on the rationality of the transmission system design. Because of the cropping patterns and agronomic requirements of soybean and wheat in the area of Huang-Huai-Hai, the time-sharing four-wheel drive HGPM chassis is designed considering the efficiency of machine field operation and field walking passability. The following basic functions are designed for the transmission system:

(1) Achieving the designed driving speed. The engine's rated speed is generally high, and the output torque is low, so it is necessary to design a reasonable transmission system and transmission ratio so that the speed of the wheels can reach the preset speed range and achieve the purpose of increasing torque.

(2) Continuously variable speed. When the plant protection machine is working, it needs to achieve stable changes of different speeds according to the changes in different environments and places to improve the utilization and adaptability of the power chassis fully.

(3) The change in direction. Most chassis engines are arranged longitudinally. That is, the direction of rotation of the engine's flywheel is in a vertical position with the forward direction of the chassis. The transmission system is needed to change the direction of power transmission to achieve the forward and reverse movement of the vehicle.

(4) Drive mode. Since the drive mode of 2WD drive and 4WD can be switched, a suitable splitter needs to be selected in the drive train to achieve this purpose.

2.2.1. Transmission Characteristics of HST

In the power chassis transmission system, the HST is a transmission machine between the diesel engine and the distributor, which is a volumetric speed control system consisting of a variable pump and a hydraulic motor, with the advantages of small size, simple operation, and easy arrangement [23,24]. The HST's input shaft is directly connected to the engine to overcome the torque required by the load driven by the output and to meet its required speed. The mechanical energy input by the engine in the form of torque and speed is converted into hydraulic energy output by the variable pump in the form of flow and pressure. Then, the variable pump inside the hydrostatic step-less transmission outputs a certain flow of pressure oil to drive the dosing motor to rotate. Then, the hydraulic energy is converted into mechanical energy. The change in variable pump displacement is achieved by controlling the swashplate tilt angle, changing the direction of the swashplate deflection and the size of the swashplate tilt angle, thus changing the flow direction and displacement of the variable pump, controlling the steering and rotation speed of the quantitative motor, and thereforeactively controlling the driving direction and driving speed of the HGPM [25].

The HGPM adopts a certain type of diesel engine to provide the driving force. Its engine-rated speed is 3000 rpm, its rated power is 12 kW, the maximum torque output is 43 N·m, and the maximum speed output is 2100 rpm. Assume that the engine output torque during operation is $M_1$ and the speed is $n_1$. In a Hydrostatic Stepless Transmission, if the losses in the piping are neglected, then the flow, torque, and power of the variable pump and the dosing motor are theoretically equal, and they all vary with the swashplate tilt angle of the variable pump [26]. The ratio of the variable pump displacement to the maximum displacement of the variables pump characterizes the degree of inclination of the swashplate of the variable pump, and its displacement ratio is expressed as

$$\varepsilon = \frac{q_p}{q_{p\max}},\tag{1}$$

where $q_p$ is the variable pump displacement, mL/r; $q_{p\max}$ is the maximum displacement of the variable pump, and mL/r; $\varepsilon$ is the variable pump displacement ratio, which can be continuously changed in the range of $-1{\sim}1$.

The output flow rates of variable pumps and quantitative motors are expressed as

$$Q_p = \varepsilon q_{p\max} n_p \eta_{pv}{}'\tag{2}$$

$$Q_m = \frac{q_m n_m}{\eta_{mv}},\tag{3}$$

where: $Q_p$ and $Q_m$ are the flow rate of the variable pump and quantitative motor, respectively, L/min; $n_p$ and $n_m$ are the speed of the variable pump and quantitative motor respectively, rpm; $\eta_{pv}$ and $\eta_{mv}$ are the volumetric efficiency of the variable pump and quantitative motor, respectively; and $q_m$ is the displacement of quantitative motor, mL/r.

Suppose the trace leakage of the connecting pipeline and the related control valve set between the hydraulic pump and the hydraulic motor is ignored. Then, it can be considered that the flow values between the hydraulic pump and the hydraulic motor in the complete hydraulic transmission unit are equal, which means $Q_p = Q_m$, and the output speed between the quantitative motors is derived as [27].

$$n_m = \frac{\varepsilon q_{p\max} n_p \eta_v}{q_m},\tag{4}$$

$$\eta_v = \eta_{pv}\eta_{mv} = \frac{1 - \frac{60\Delta p C_s}{\mu n_p \varepsilon}}{1 + \frac{60\Delta p C_s}{\mu n_m}},\tag{5}$$

$$\eta_H = \eta_p \eta_m = \frac{\left(1 - \frac{60\Delta p C_s}{\mu n_p \varepsilon}\right)\left(1 - C_f - \frac{C_v \mu n_p \varepsilon}{60\Delta p}\right)}{\left(1 + \frac{60\Delta p C_s}{\mu n_m}\right)\left(1 + \frac{C_f}{\varepsilon} + \frac{C_v n_p \varepsilon}{60\Delta p}\right)},\tag{6}$$

$$n_m = \frac{q_{p\max} n_p \varepsilon}{q_m} - \frac{q_{p\max} 60\Delta p C_s}{q_m \mu} - \frac{60\Delta p C_s}{\mu},\tag{7}$$

where $\eta_v$ is the total volumetric efficiency of the hydrostatic transmission; $\eta_H$ is the total efficiency of the hydrostatic transmission; $\eta_p$ is the total efficiency of the variable pump; $\eta_m$ is the total efficiency of the quantitative motor; $\Delta p$ is the pressure difference between the high and low-pressure side, MPa; $Cs$ is the laminar flow coefficient, $Cs = 9 \times 10^{-10}$; $\mu$ is the hydraulic oil dynamic viscosity, Pa·s.

The torque output of a fixed motor is expressed as

$$M_m = 159 q_m \Delta p \eta_{mm},\tag{8}$$

$$\eta_{mm} = 1 - C_f - \frac{C_v \mu n_m}{60\Delta p},\tag{9}$$

where $\eta_{mm}$ is the mechanical efficiency of the quantitative motor; $C_f$ is the mechanical resistance coefficient, $C_f = 1 \times 10^{-2}$; $C_v$ is the laminar flow resistance coefficient, $C_v = 2 \times 10^5$; the coefficient 159 is the quantitative conversion value of the rotational radian, numerically equal to $1000/(2\pi)$.

Due to the use of a synchronous belt between the diesel engine and the variable pump to transfer power, which means $n_p = n_1$, the output torque and power of the quantitative motor can be expressed as follows:

$$M_m = 159 q_m \left[\Delta p - \Delta p C_f - \frac{\mu C_v}{60}\left(\frac{q_{p\max} n_1 \varepsilon}{q_m} - \frac{q_{p\max} 60\Delta p C_s}{q_m \mu} - \frac{60\Delta p C_s}{\mu}\right)\right],\tag{10}$$

$$\begin{aligned}P_m &= \frac{M_m n_m}{9550}\\ &= \left[\frac{\left(q_m \Delta p - C_f q_m \Delta p + q_{p\max} \Delta p C_s C_v + q_m \Delta p C_s C_v\right)}{60} - \frac{\mu q_{p\max} n_1 \varepsilon C_v}{3600}\right]\left(\frac{q_{p\max}(\mu n_1 \varepsilon - 60\Delta p C_s)}{q_m \mu} - \frac{60\Delta p C_s}{\mu}\right)\end{aligned}\tag{11}$$

From Equations (6) and (11), it can be seen that the power transfer characteristics of the hydrostatic continuously variable transmission are related to the displacement ratio of the variable pump and the pressure difference between the high- and low-pressure sides. When the displacement ratio $\varepsilon$ varies in the range from 0 to 1, there is a certain range of dead zone in the output speed of the dosing motor as the displacement ratio varies due to the leakage phenomenon during the operation of HST. $\varepsilon = 0$ means that the variable pump swashplate is in the neutral position. At this time, the hydrostatic step-less transmission does not output power to the outside, and the state can be used for the idle speed of the HGPM, gear shifting, braking, and other working conditions. At displacement ratio $\varepsilon > 0.32$, HST is in the high-efficiency zone with efficiency above 58%, and this displacement ratio range applies to the normal operating conditions of the high clearance self-propelled sprayer. For $\varepsilon < 0.32$, the small displacement zone is only applicable to excessive operating conditions, such as starting and braking of the high clearance self-propelled sprayer.

### 2.2.2. Transmission Characteristics of the Splitter

In the choice of drive form, compared to two-wheel drive, four-wheel drive power chassis has higher scene adaptability, can adapt to a wider range of terrain, and in the plains and shallow hilly areas can offer better performance. Four-wheel drive has greater traction and lowers slip losses than two-wheel drive, resulting in easy driving on sticky, wet, and sandy soils, good off-road performance, improved traction adhesion, and good passing power. The disadvantage is that the structure is more complex [28–30]. After comprehensive analysis and comparison, the HGPM adopts a time-sharing four-wheel

drive splitter model, which selects the 4WD mode when the operation requires high traction and switches to the 2WD mode when the road is shifted, or the road is dry and flat.

Inside the splitter, there are two sets of switching mechanisms: the high-low gear (H-L) gear switching structure and the two-four drive (2-4) mode switching structure. H-L switching mechanism is responsible for switching the gear ratio of the splitter. It consists of only one engagement sleeve, without synchronizing mechanism, which can form three gears when in different positions, namely high speed (H), neutral (N), and low speed (L). The 2-4 mode switching mechanism contains a synchronization mechanism responsible for switching the vehicle's drive mode, which can form two modes, 2WD and 4WD, respectively. The HGPM can form two driving modes and three driving states by two switching mechanisms cooperating. The transmission ratios for different drive modes and drive states are shown in Table 1.

**Table 1.** Transmission ratios for different drive modes and drive states.

| Drive Mode | Gear Level | Transmission Ratio | Drive Form |
|---|---|---|---|
| 2WD | 2H | 1:1 | Two-wheel-drive status |
| 4WD | 4H | 1:1 | Four-wheel high-speed drive status |
|  | 4L | 2.48:1 | Four-wheel low-speed drive status |

According to theoretical calculations and a review of related studies, the 4L mode can provide greater tractive force to the whole machine than the 4H mode [31], so in this paper, the 4H mode is used to study the off-road performance of upland gap planters under 4WD conditions.

### 2.3. System Configuration

The power transmission system of HGPM adopts a combination of hydraulic and mechanical transmission modes, which can realize step-less variable speed and meet the requirements of body leveling, plant protection spraying, fertilizing, and other working parts. Combined with the different operating conditions, the power transmission routes of the upland gap planters under each driving state are shown in Figure 2.

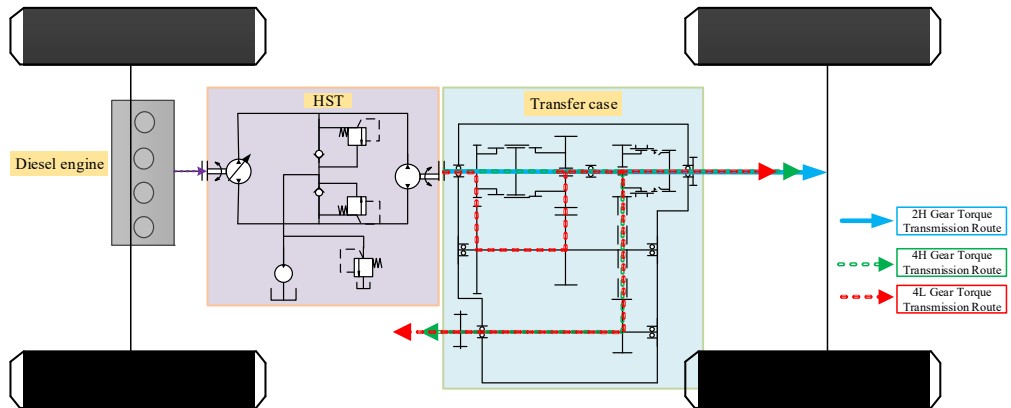

**Figure 2.** Chassis drive train diagram.

To regulate the engine operating range and reduce the wide range of torque fluctuations caused by load shocks, a hydrostatic transmission is used as the power regulation unit, and a splitter is used as the power transmission unit. The whole power output structure can be coupled and shunted with the splitter and engine through the hydrostatic transmission to achieve full coverage of HGPM power in multiple operating modes through different combinations [32]. To facilitate the study of the torque transfer route, this section focuses on establishing the torque transfer equations for different gears of the power-coupled system.

When the HGPM is in 2H gear, the active synchronization part is subject to the engine's dynamic moment and the road's resistance moment. According to Newton's second law, when the vehicle is in 2WD, the dynamics of the vehicle drivetrain are modelled as follows:

$$T_c - T_L = J_r \frac{d\omega_r}{dt},$$
(12)

$$T_L = \frac{\left(T_f + T_j + T_w\right)}{i_0},$$
(13)

where $T_c$ is the torque transferred from the engine to the rear driveshaft, N·m; $T_L$ is the equivalent driving resistance moment to the rear driveshaft, N·m; $i_0$ is the front/rear main reduction ratio, 4.1; $T_f$ is the rolling resistance moment, N·m; $T_j$ is the acceleration resistance moment, N·m; $T_w$ is the air resistance moment, N·m; $\omega_r$ is the angular velocity of the rear driveshaft, rad/s.

When the vehicle is driving at a constant speed, the angular velocity $\omega_r$ of the rear drive shaft is a constant value, so the torque output by the distributor at this time is balanced with the driving resistance torque, i.e.,

$$M = M_m = 159 q_m \left[ \Delta p - \Delta p C_f - \frac{\mu C_v}{60} \left( \frac{q_{p\max} n_p \varepsilon}{q_m} - \frac{q_{p\max} 60 \Delta p C_s}{q_m \mu} - \frac{60 \Delta p C_s}{\mu} \right) \right]$$
(14)

When the HGPM is in 4H mode, the splitter divides the torque transferred from the engine to the front and rear driveshafts into $M_F$ and $M_R$ under the working condition of the vehicle driving at a constant speed. According to the ratio of the splitter in 4H mode, the dynamics of the vehicle drivetrain at this time can be modelled as follows:

$$M_F = M_R = \frac{1}{2} M_m = 79.5 q_m \left[ \Delta p - \Delta p C_f - \frac{\mu C_v}{60} \left( \frac{q_{p\max} n_p \varepsilon}{q_m} - \frac{q_{p\max} 60 \Delta p C_s}{q_m \mu} - \frac{60 \Delta p C_s}{\mu} \right) \right]$$
(15)

When the HGPM is in 4L mode, the splitter divides the torque transferred from the engine to the front and rear driveshafts into $M_F$ and $M_R$ under the working condition of the vehicle driving at a constant speed. According to the ratio of the splitter in 4L mode, the dynamics of the vehicle drivetrain at this time can be modelled as follows:

$$M_F = M_R = 1.24 M_m = 197 q_m \left[ \Delta p - \Delta p C_f - \frac{\mu C_v}{60} \left( \frac{q_{p\max} n_p \varepsilon}{q_m} - \frac{q_{p\max} 60 \Delta p C_s}{q_m \mu} - \frac{60 \Delta p C_s}{\mu} \right) \right]$$
(16)

Equations (14) and (16) show that the power transmission system of the upland gap planters is related to the displacement ratio of the hydrostatic transmission and the engine output speed. Since the variable pump and quantitative motor are used, the speed-torque ratio of the whole system can be controlled by adjusting the variable pump displacement.

## 3. Chassis Performance Simulation Analysis

### 3.1. Dynamics Modelling

Simulation model creation is based on SolidWorks software to create 3D solid models and export them in Parasolid (*.x-t) format. The model of the whole machine is established, and the main parameters are shown in Table 2.

Then, the model is imported into RecurDyn software to maintain the assembly relationships and simplify the solid model [33,34]. Due to the simplification of the model, the weight of the machine differs greatly from the actual weight, so it is reset according to the actual weight of the machine.

**Table 2.** The main parameters of modeling.

| Parameters | Number of Values |
|---|---|
| Full load mass | 1500 kg |
| Overall dimension | 2200 mm × 1400 mm × 1800 mm |
| Spraying width | 8000 mm |
| The volume of medicine box | 200 L |
| Operating speed | 0~10 km/h |
| Unloaded mass | 1200 kg |
| Number of axes | 2 |
| Axis distance | 1200 mm |
| Wheel distance | 1650 mm |
| Distance from center of mass to front axis | 800 mm |
| Ground clearance | 980 mm |

To reduce the simulation computing time, the following simplifications were made to the structure of the plant protection machine without affecting the simulation tests:

(1) All wheel mass properties and geometric dimensions are the same.

(2) Except for the damping, elasticity, and rubber component tires, the remaining parts are considered rigid bodies and no deformation is considered.

(3) The spring-loaded mass is considered a rigid body, with appropriate simplification for the connection flexibility between the rigid bodies.

(4) The internal friction of the moving parts is neglected.

(5) The non-major transmission parts are simplified, and the threaded nut, bearing, etc., replaced with fixed and rotating ones.

(6) The impact of the suspension system on the whole machine is ignored.

The wheel model is set up in the Tire module of RecurDyn, the tire–ground contact force is set, and the tire parameters of the tire mechanics model are customized according to the shape difference between the planters' tire model and the default Tire Group model [31].

The SolidWorks software is used to establish a 3D model of the corresponding working condition pavement, and the change in tire force under different soil conditions is achieved by changing the dynamic friction coefficient in the contact parameters between the pavement and the tire. We add solid contact between the tires and the ground and between the gears of the differential and add rotating subsets between the wheels and the frame, the front and rear drive shafts, and the differential output shafts through the Joint command. Based on the transmission relationship between the plant protection machine systems and the corresponding constrained connections to form the whole machine model, the multi-case performance simulation test was conducted, as shown in Figure 3.

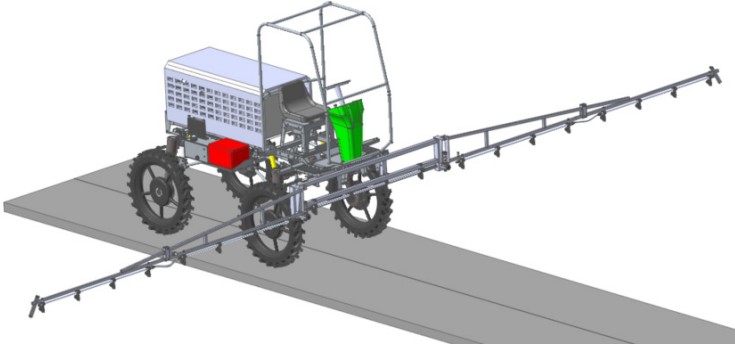

**Figure 3.** Whole machine model.

*3.2. Chassis Performance Analysis*

3.2.1. Climbing Performance Analysis

There is a limit slope angle $\alpha$ in the chassis climbing process; when the slope angle exceeds this limit value, the chassis are unable to pass. Due to the constant-torque speed

regulation characteristics of the hydrostatic transmission, the speed is selected as the power input to the input axis in the multi-body dynamics simulation in this paper. The velocity drive function is applied to the front output axis rotation sub of the HGPM, and the corresponding drive function is STEP (TIME, 0, 0, 0.1, 30); the simulation time is set at 25 s and simulation step is set at 350. The climbing test of HGPM in the 2H and 4H modes was carried out, respectively, as shown in Figure 4.

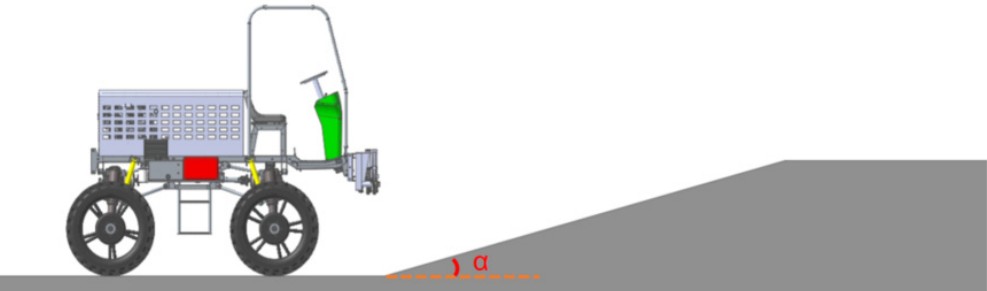

**Figure 4.** Climbing model of HGPM.

For most agricultural lands, the maximum slope angle does not exceed 25° [35]. In this paper, the road slope is set to 25°, 30°, 35°, and 40°, respectively, and the body pitch angle and climbing speed are shown in Figure 5.

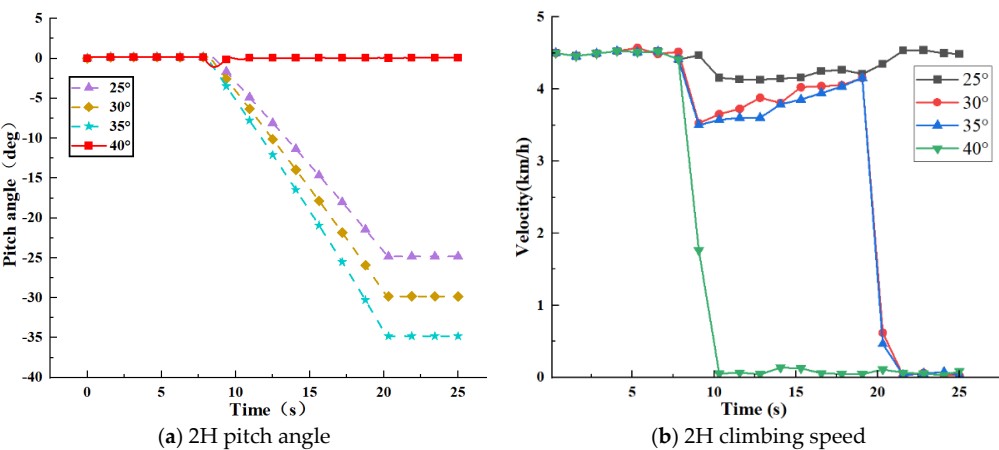

(**a**) 2H pitch angle                                              (**b**) 2H climbing speed

**Figure 5.** Results of the 2H climbing test.

From the simulation data curves in Figure 5a,b, it can be seen that in the HGPM in 2WD mode, the 0−9 s range is the smooth start to the uniform speed driving phase, and the 9−20 s range is the climbing phase. When the slope is at 25°, the 9−10 s speed is reduced to 4.16 km/h for the initial stage of climbing, the 10−19 s speed is stabilized at approximately 4.16 km/h for the uniform climbing stage, the 19−20 s speed is gradually returned to the initial state for the end of climbing, the 20−25 s speed is basically stabilized for the completion of the climbing and uniform driving stage. With the change in time, the planters start to climb the slope, body speed lowered first, driving at a constant speed on the ramp for a period of time into the end of the climb; that is, the front end climbing to the top of the slope, followed by the body speed slowly decreasing to the initial value. When the slope is at 30° and 35°, the HGPM is in the 0−9 s range for a smooth start to the uniform speed driving stage. In the 9−10 s range, the vehicle speed is rapidly reduced to a peak of 3.5 km/h. In the 10−19 s range, the vehicle gradually begins to shake, the speed entering a fluctuating state. In the 19−20 s range, vehicle speed drops sharply. In 20−21 s range, vehicle speed gradually becomes 0. Currently, the driving force is not enough to support the plant protection machine forward. When the slope is at 40°, due to the high slope, the vehicle speed drops to 0 rapidly in the 9 to 10 s range, and the vehicle

speed remains constant in the 10 to 25 s range because the driving force cannot support the driving resistance of the plant protection machine. The whole 2H climbing process can be observed, in the same speed conditions, body speed decreasing with the increase in the slope steepness. The steeper the slope, the faster the speed drop, and the slower the climbing speed. When the driving force is not sufficient to support the upward traction, the body speed will first drop sharply and then keep gradually dropping to 0; then, the planters will fail to climb the slope. However, when the driving force meets the demand, the body speed will first reduce to the lowest value, then complete the climbing stage at a uniform speed, and the speed will gradually return to the initial value at the end of the climb; in this case, the climb will be successful. In the two-wheel drive mode, the chassis of the upland gap planters decreases with the increase in body speed at the slope of 25°, and the body speed remains unchanged after the 20 s mark, indicating that the planters have reached the top of the slope at this time. The climbing speed gradually decreases in the process of slope climbing from 30° to 40°, and finally amounts to 0, which indicates that the planters cannot move up the slope normally in this slope range.

From the simulation data curves in Figure 6a,b, it can be seen that for the HGPM in 4H mode, the 0−8 s range is the smooth start to the uniform speed driving phase, and the 8−20 s range is the climbing phase. When the slope is at 25°, the 8−9 s speed decreases to 4.9 km/h for the initial stage of climbing, the 9−19 s speed stabilizes at approximately 5 km/h for the uniform climbing stage, and at 19−20 s, the speed starts to rise gradually back to the initial state. In the 20−25 s range, the speed stabilizes for the completion of climbing and uniform driving stage. With the change in time, the planters start to climb the slope, body speed lowered first, driving at a constant speed on the ramp for a while into the end of the climb; that is, the front end climbing to the top of the slope, followed by the body speed slowly decreasing to the initial value. When the slope is at 30°, the 0−8 s range is the stage of smooth start to uniform speed; the 8−9 s speed rapidly decreases to peak 4.6 km/h; the 9−19 s speed is in dynamic equilibrium, where the planters gradually adapt to the climbing condition and start the uniform climbing stage. The speed starts to rise and gradually returns to the initial state, which is the end of the climbing tail section. The speed is stabilized at the 20−25 s range, which is the end of the climbing and uniform speed stage. When the slope is at 35°, the 0−8 s range is the stage of smooth start to uniform speed, the 8−9 s speed rapidly decreases to reach the peak of 4.6 km/h, the 9−19 s speed is in dynamic equilibrium, where the planters gradually adapt to the climbing condition and start to climb at uniform speed and the speed starts to rise and gradually returns to the initial state, which is the end of the climbing tail section. In the 20−25 s range, the speed is stabilized, which is the stage of completion of climbing and uniform speed. The speed is stabilized in the 20−25 s range, which is the end of the climbing and uniform speed stage. When the slope is at 40°, due to the height of the slope, the vehicle speed drops rapidly to 4.24 km/h in the range from 8 to 9 s, the vehicle speed is in dynamic equilibrium in the range from 9 to 19 s, and the plant protection machine starts the difficult climbing process. As the slope height increases, the gravitational force of the planters along the vertical direction of the ground, the interaction force between the driving wheels and the ground decreases, and the driving force cannot support the driving resistance of the planters leading to a sharp drop in speed to 0 from 19 to 25 s, indicating that the 4H mode upland gap planters cannot climb over the slope of 40°, and the maximum climbing degree of the upland gap planters in the 4H mode is at least 35°. From the whole 4H hill-climbing process, it can be seen that the driving torque required for hill climbing is approximately proportional to the slope angle, i.e., the higher the slope, the higher the driving torque required. In the same speed conditions, body speed decreases with the increase in the slope steepness. The steeper the slope, the faster the speed drop, and the slower the climbing speed. When the driving force is not sufficient to support the upward traction, the body speed will first drop sharply, and the continue to gradually drop to 0. The plant protection machine will fail to climb the slope. At the same time, when the driving force meets the demand, the body speed will first reduce to a minimum value and then complete the climbing stage at a uniform

speed, and the speed will gradually return to the initial value at the end of the climb; in this case, the climb will be successful. Within the slope steepness range of $0-35°$, the speed of the HGPM chassis under the 4H mode decreases with the increase in the slope steepness, eventually climbing to the top. The speed of the HGPM gradually decreases to 0 at the slope steepness of $40°$, indicating that the upland gap planters in 4H mode cannot climb over the slope of $40°$. The maximum climbing degree of the HGPM in 4H mode is at least $35°$.

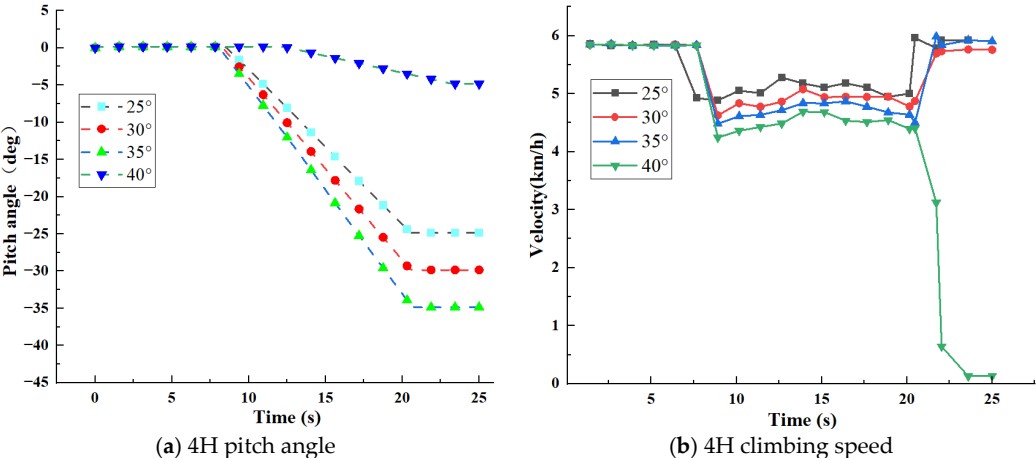

**Figure 6.** Results of the 4H climbing test.

3.2.2. Cross-Ridge Performance Analysis

The speed driving function is applied to the front output axis rotation sub of the HGPM. The corresponding driving function is STEP (TIME, 0, 0, 0.1, 30), the simulation time is set at 25 s, simulation step set at 350. The HGPM ridge crossing test is carried out in 2H and 4H mode, respectively, as shown in Figure 7.

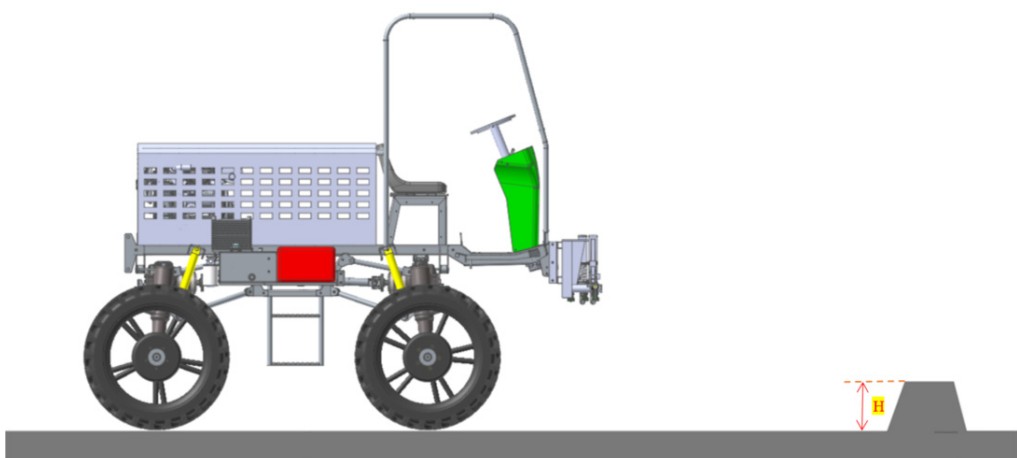

**Figure 7.** Ridge crossing simulation model of HGPM.

The 2H mode has the following four different working conditions: the height of the ridge is set to 50 mm, 100 mm, 150 mm, and 200 mm, respectively. The simulation results are shown in Figure 8a,b.

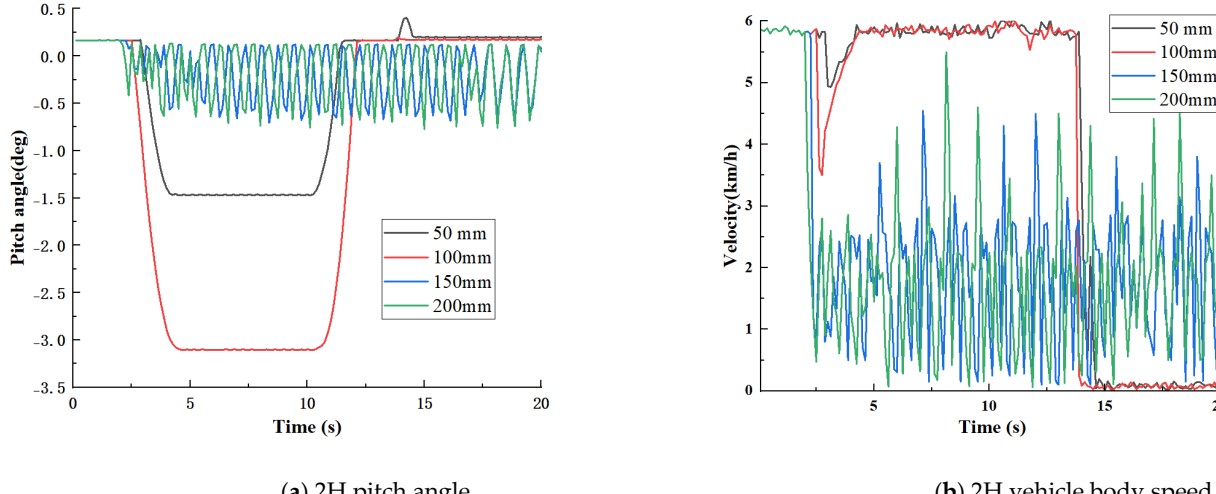

(**a**) 2H pitch angle        (**b**) 2H vehicle body speed

**Figure 8.** Simulation results of 2H ridge crossing.

After the simulation is stabilized, the velocity of the vehicle body and the change curve of the chassis pitch angle under the 0−20 s HGPM chassis over the ridge working condition are extracted, as shown in Figure 8a,b. In 2H mode, the pitch angle starts to drop from 3 to 4.5 s; the pitch angle is stable and unchanged from 4.5 to 10.5 s. At a constant speed, the process of crossing the ridge occurs. The body attitude gradually returns to the right stage when crossing the ridge from 10.5 to 12 s; from 12 to 20 s, the plant protection machine passes the ridge and travels at a constant speed. The speed of the vehicle drops sharply within 3−4.5 s at the beginning of the ridge crossing phase and then stabilizes until the whole process of ridge crossing is completed. At a ridge height of 50 mm, the HGPM could easily pass through, and the speed drops inconspicuously and quickly stabilizes, and then the vehicle speed is maintained at 5.5 km/h at a constant speed to complete the ridge crossing. At the ridge height of 100 mm, 3−4.5 s is the start of the ridge crossing phase. The pitch angle drops to the slope value, the body speed drops to 3 km/h, and then quickly returns to the steady state of 5.5 km/h, starting the uniform speed driving phase. At the ridge heights of 150 mm and 200 mm, the body pitch angle is always shaking, and the vehicle speed is always unstable after the first drop, indicating that the planters could not pass the ridge of that height at this time. The test results showed that the HGPM could pass the ridge smoothly when the ridge height was 50 mm and 100 mm, and the speed of the front wheels crossing the ridge at 100 mm ridge height dropped by more than 50 mm, while at 150 mm and 200 mm ridge height, the HGPM were always shaking and could not pass the ridge.

There are seven different working conditions in 4H mode: the height of the ridge is set to 150 mm, 200 mm, 250 mm, 300 mm, 320 mm, and 330 mm, respectively. The simulation results are shown in Figure 9a,b.

After the simulation is stabilized, the body speed and chassis pitch angle change curves under 0−20 s upland gap planters chassis over the ridge working condition are extracted, as shown in Figure 9a,b. In the 4H mode, the pitch angle of the body starts to drop from 3 to 4.5 s, the pitch angle is stable and unchanged from 4.5 to 10.5 s, and the process of crossing the ridge at a constant speed occurs. From 10.5 to 12 s, the stage of the front wheels crossing the ridge occurs, and the body attitude gradually returns to positive; from 12 to 20 s, there is the stage of the rear wheels of the plant protection machine passing the ridge and driving at a constant speed. The speed of the vehicle drops sharply within 3−4.5 s at the beginning of the ridge crossing phase and then stabilizes until the whole process of ridge crossing is completed. At the ridge height of 150 mm, 200 mm, 250 mm, 300 mm, 320 mm, 3−4.5 s is the start of the ridge crossing phase. The pitch angle drops to the slope value, the body speed drops and then quickly returns to the steady state of 5.5 km/h, starting the uniform speed driving phase. While the height of the ridge is 330 mm, although the front wheels pass the ridge at the ridge height of 330 mm, the rear

wheels does not pass completely, indicating that the maximum ridge crossing height of the 4H mode upland gap planters is between 320 mm and 330 mm. The test results show that in 4H mode, the HGPM maintains good passability when the ridge height is 100 mm to 320 mm, and the speed drop becomes larger as the ridge height increases. Although the front wheels pass the ridge at the ridge height of 330 mm, the rear wheels do not pass completely, indicating that the maximum ridge crossing the height of the 4H mode HGPM is between 320 mm and 330 mm.

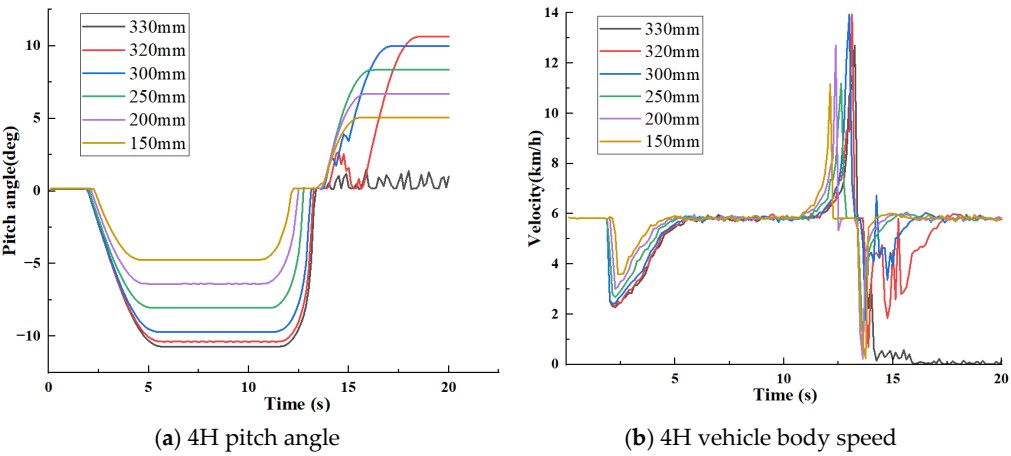

(**a**) 4H pitch angle        (**b**) 4H vehicle body speed

**Figure 9.** Simulation results of 4H ridge crossing.

### 3.2.3. Performance Analysis of Opposite Pavement

This section designs the simulation conditions of HGPM under high attachment and low attachment opposing road surfaces. The coefficient of dynamic friction between tires and ground is set to 0.8 and 0.3 to simulate the vehicle driving on opposing road surface environments. The speed driving function is applied to the front output axis rotation sub of HGPM; the corresponding driving function is STEP (TIME, 0, 0, 0.1, 10), the simulation time is set to 3.5 s, and the simulation step is set to 100. To carry out the simulation, working condition experiments of accelerating sharply on the variable attachment road of HGPM in 2H mode and 4H mode, respectively, are performed as shown in Figure 10.

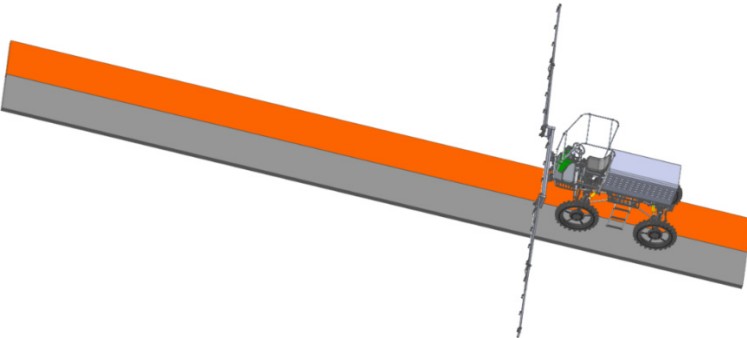

**Figure 10.** Simulation model of opposed pavement.

The driving force provided by the ground to the vehicle is the guarantee of the driving performance of the whole vehicle. Generally speaking, the greater the driving force, the better the power performance of the whole vehicle. The size of the driving force is closely related to the adhesion conditions of the road. Under the 2H mode and 4H mode, respectively, the simulation test of rapid acceleration driving on variable attachment roads was carried out, and the adhesion coefficient of the left side road was set to 0.3, and the right side road was set to 0.8. The simulation results are shown in Figures 11 and 12.

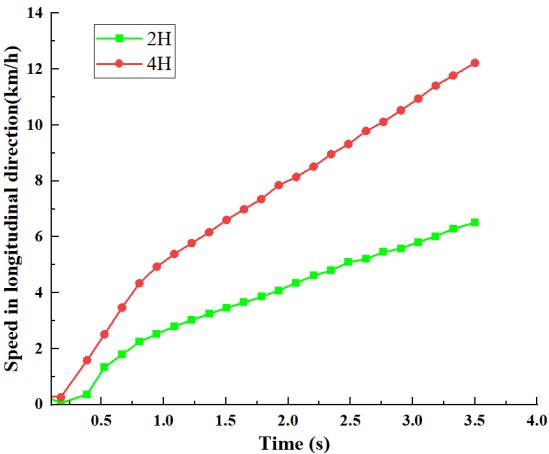

**Figure 11.** Result of longitudinal vehicle speed simulation.

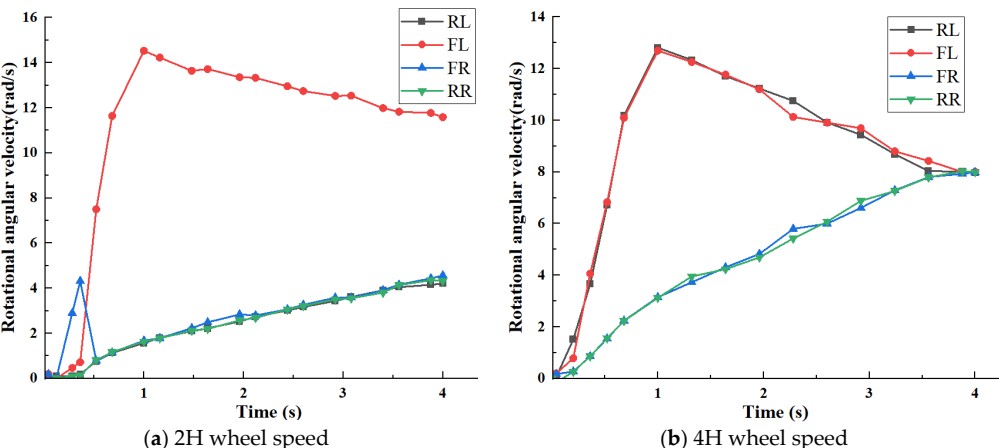

(**a**) 2H wheel speed                (**b**) 4H wheel speed

**Figure 12.** Simulation test results of opposed pavement.

After simulation stabilization, the body speed variation curve under 0−4 s upland gap planter chassis over the ridge condition was extracted, as shown in Figure 11. Simulation experimental results show that with the 0−0.17 s car speed in 2H mode, the acceleration is too rapid, and the road surface skids swiftly, resulting in the body speed not increasing, but decreasing. In the 0.17−0.9 s range, body speed begins to increase to 2.5 km/h faster; in the 0.9−4 s range, due to increasing speed, wheel skid intensifies, resulting in slow growth of vehicle driving speed, until the maximum speed of 6.5 km/h. In 4H mode, the 0−0.17 s car speed, also due to the rapid acceleration situation, the road surface skids intensely, causing the body speed not to increase but to decrease; in the 0.17−0.9 s range, body speed began to increase to 4.9 km/h faster; in the 0.9−4 s range, body speed wheel skid intensified due to the increasing speed, resulting in slow growth of vehicle driving speed until the maximum speed of 12.2 km/h. The speed maxima of 2H and 4H modes are 6.5 km/h and 12.2 km/h, respectively, with significant differences in short-time speedups, which indicates that the changes in drive modes of the HGPM have a significant impact on their acceleration characteristics when driving on open roads.

After the simulation is stabilized, the body speed variation curves under 0−4 s upland gap planters chassis over the ridge conditions are extracted, as shown in Figure 12a,b. The test results show that as the left side of the road is low attachment and the right side is high attachment, in 2H mode, the HGPM is in the front-drive state, and the left front wheel is the driving wheel. The 0−0.4 s range rotational angular speed increases slowly to 0.78 rad/s, the 0.4−1 s range rotational angular speed increases rapidly to the peak 14.5 rad/s, the 1−4 s range wheel speed starts to stabilize slowly and converges toward the steady state of the left and right wheel speed. The left rear wheel is the driven wheel, the 0−0.4 s

range rotational angular speed with the dynamic value adds a wave peak of 4.3 rad/s, the 0.4−0.5 s range drops sharply to 0.8 rad/s, the 0.5−4 s range wheel speed with the left front wheel and left rear wheel increases amplitude, rate maintained at the same level. When the right front wheel and right rear wheel drive on the high adhesion road and the road surface has good adhesion conditions, the road surface can provide the driving force required by the whole vehicle, and the adhesion force of the road surface can be fully utilized at this time. The wheel speed slowly increases to the maximum value of 4.5 rad/s, and the left and right wheel speeds tend to converge, but the convergence is not completed in the 0−4 s range to return to the steady state. In 4H mode, the left rear wheel and the left front wheel remain the same. From 0 to 1 s, the rotational angular speed increases rapidly to the maximum value of 12.6 rad/s, and from 1 to 4 s, the wheel speed gradually returns to the stable value of 8 rad/s. The changes in the right front wheel and the right rear wheel speed remain the same, and from 0 to 4 s, the wheel speed always increases slowly to the maximum value of 8 rad/s and then remains in the dynamic stable state after 3.5 s. The rotational angular speeds of the left and right wheels converge, and the rotational angular speed remains the same at 8 rad/s. From the above analysis, it can be seen that the angular speed of rotation of the coaxial left low attached wheel is significantly greater than that of the right high attached wheel due to the vehicle acceleration and the backward shift of the vertical load. As the speed increases, the driving force gradually decreases until the right tire adhesion limit is less than 3.5 s, and the speed of each wheel converges to 8 rad/s smoothly, indicating that the 4H mode effectively avoids excessive wheel slip and improves the lateral stability of the vehicle. When the road surface adhesion conditions are poor, the wheels and the road surface cannot maintain a good adhesion relationship. If the wheel slips, the ground can cause the wheel force to significantly reduce. Therefore, it can be seen that the vehicle's power performance is related not only to the size of the driving torque but also directly to the adhesion relationship between the tires and the ground.

## 4. Experiments

### 4.1. Climbing and Ridge Crossing Performance Test

To further verify the rationality of the simulation results, performance experiments of the trial upland gap planters under two operating environments were conducted in this section at the Electromechanical Engineering Park of Anhui Agricultural University (longitude: 117.257757, latitude: 31.860520). The ridge crossing and hill climbing tests are conducted mainly for the passing performance of the chassis, and the test data results are used to determine whether the requirements are met. Due to the limitation of the site, the climbing test was only carried out on a fixed ramp with a slope $\theta$ of 15°, as shown in Figure 13.

The HGPM was carried out to pass the slope test in 2H mode and 4H mode, respectively, and the pitch angle and speed of the body and the time used in the uphill process were recorded, as shown in Figure 14.

As shown in Figure 14a,b, under the specific ramp in the 15° working condition, the 0−9 s range is the smooth start to the uniform speed phase, and the 9−20 s range is the climbing phase. Since the plant protection machine will have some vibration or sensor accuracy when driving on the ground or on the ramp, the body pitch angle will also have some jitters, but the overall curve is similar to that of the simulation results. At the same time, with the time change, the body speed in the climbing stage is first lower than the uniform speed; in the 9−20 s climbing stage, the speeds were kept at 5.3 km/h, 4.8 km/h or so approximate uniform speed. The test results show that the 4H mode has a larger speed difference compared to the 2H mode, which reflects the need for variable speed and increased torque, and improves the passing performance. The test results are all consistent with the previous simulation results.

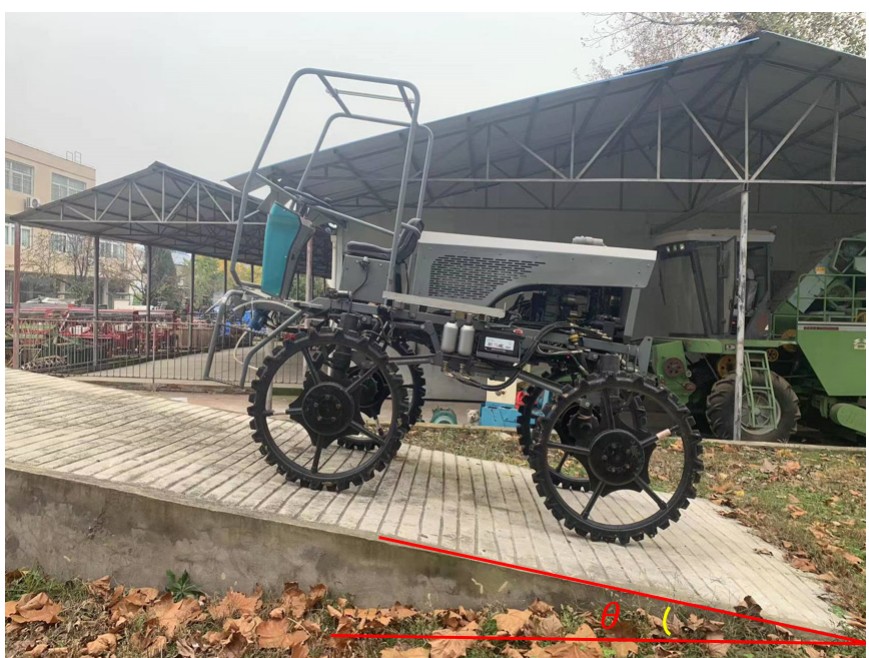

**Figure 13.** The climbing experiment of HGPM.

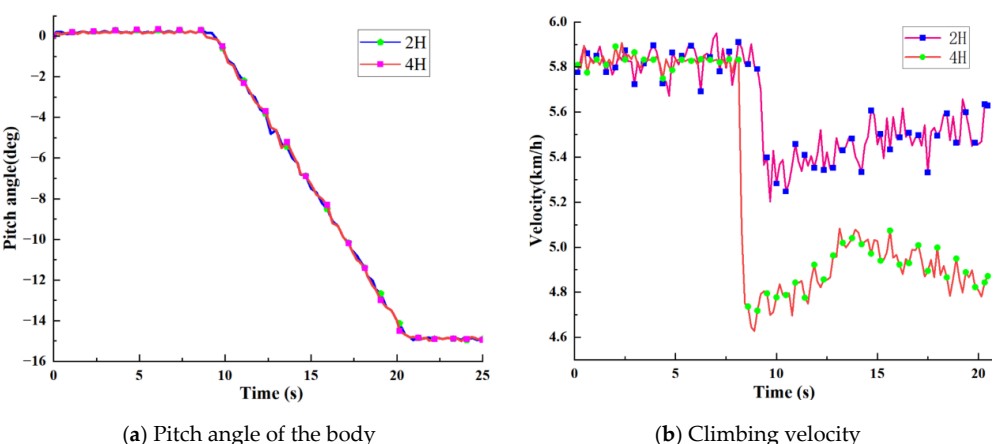

(**a**) Pitch angle of the body

(**b**) Climbing velocity

**Figure 14.** Climbing experiment results.

For the ridge crossing experiment, due to the limitation of the site, a field ridge with height H and slope β was simulated manually for the test, and HGPM was carried out in 2H mode and 4H mode to cross the ridge of different heights at a constant speed. The pitch angle, speed and time spent in the uphill process were recorded. The test was conducted by manually simulating the ridge of different heights, and the test data at a ridge height of 320 mm were extracted for analysis. The test results are shown in Figure 15.

As shown in Figure 15a, when the HGPM crossed the ridge in 2H mode, under the working condition of ridge height H of 320 mm and slope β of 26°, the pitch angle of the body was always fluctuating, and the speed decreased in the 2.8−3.2 s range, slowly increased to the initial speed in the 3.2−4.3 s range, and then dropped sharply to 0 after 13.5 s. This demonstrated that the planters could not pass the 320 mm ridge in the 2H mode. As shown in Figure 15b, under the same test conditions, when the ridge crossing was carried out in 4H mode, the pitch angle first dropped to the slope value, and then the ridge crossing test was completed. The body speed dropped at the beginning of the ridge crossing phase and then quickly returned to the steady state of 5.5 km/h and started the uniform speed driving phase. The experimental results show that the 4H mode can pass the ridge with a ridge height of 320 mm and a slope of 26° smoothly, while the 2H

mode will display a short fluctuation of speed, dropping sharply to 0. The test results are all consistent with the previous simulation results. The performance test of the prototype verified the rationality and authenticity of the data, such as theoretical design and related dynamics analysis of the HGPM.

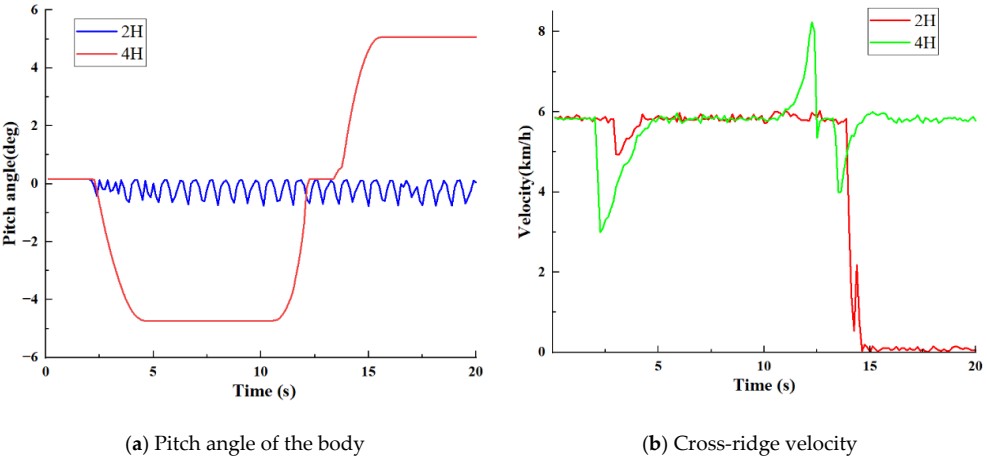

(**a**) Pitch angle of the body      (**b**) Cross-ridge velocity

**Figure 15.** Results of the ridge crossing test.

### 4.2. Real-World Test of Rapid Acceleration on Variable-Access Roads

In this section, the real vehicle acceleration test on variable adhesion pavement is designed to verify whether the simulation results meet the agreement under the real variable adhesion pavement, so a test field with similar soil characteristics to those in the Huang-Huai-Hai region is selected for the test, and the soil of the test field is treated to obtain a working surface similar to the simulation to verify the validity of the simulation test results [36]. The HGPM is driven in 2H mode and 4H mode to conduct the test on the variable attachment folio field surface, keeping the throttle open until the whole machine passes the test surface. The vehicle's driving data are collected using the tilt and wheel speed sensors, and the experiment scenario is shown in Figure 16.

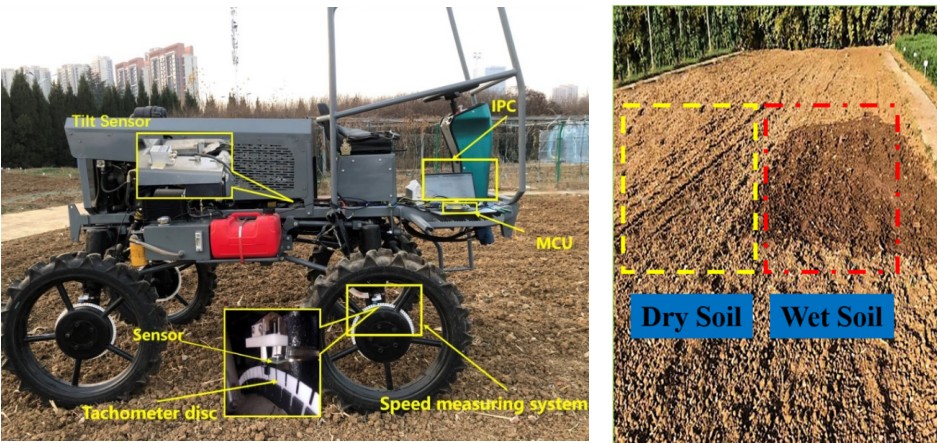

**Figure 16.** Opposite pavement experiment.

After the test was completed, the data obtained from each data acquisition module in the upper computer were processed separately to obtain the body speed curve, as shown in Figure 17, and the wheel speed variation curve, as shown in Figure 18.

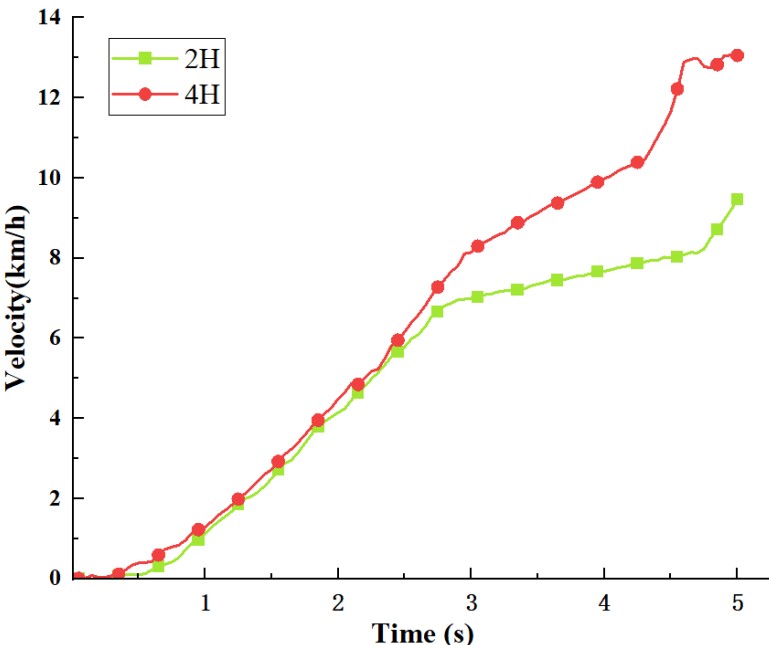

**Figure 17.** Opposite road test speed data.

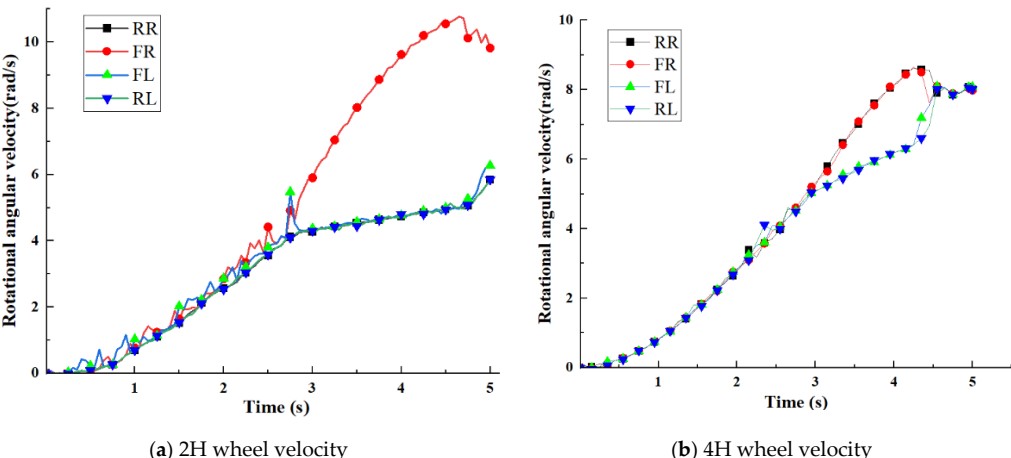

(**a**) 2H wheel velocity       (**b**) 4H wheel velocity

**Figure 18.** Opposite road test wheel speed data.

As shown in Figure 17, at the beginning of the test, since the plant protection machine was driven by the front wheels in 2H mode when driving to the wet road, the front wheels entering the wet soil first began to slip, and the body speed began to drop, and when the front wheels exited the wet soil, the speed quickly stabilized and showed an upward trend. Since both front and rear axles have the drive in 4H mode, when the front wheels slip, the rear wheels can still provide forward power, only partially losing motion. The test results show that the acceleration of the whole machine in 4H mode is significantly higher than that in 2H mode on the wet road.

As shown in Figure 18a,b, during the test, the output speed of the right low attached wheel in 4H mode increases in the 3−4.5 s range and then converges rapidly. In 2H mode, the right front wheel speed rises rapidly to a high value within the 2.8−4.7 s range when it enters the wet road with low attachment, while the left front wheel speed at high attachment remains stable and continues to deliver power. The test results show that changing the drive mode and adjusting the front and rear axle output torque using dynamics control can effectively avoid excessive wheel slip and quickly pass through the skidding zone.

## 5. Discussion

(1)    The influence of dynamics control on operational performance

During the operation of HGPM, the sudden change in the ground type is very likely to cause the inconsistency of the four wheels' speed and produce the phenomenon of one or two wheels slipping, and the sudden change in the ground type mainly includes ridge, slope, pit, deep mud foot, etc. In addition, when climbing slopes and crossing ridges, the lack of traction causes the two front wheels to slip, resulting in a lack of power in front-drive vehicles and affecting the passability of agricultural machinery. 2H and 4H are freely switchable for power control to achieve the best power performance. With acceleration performance, hill climbing, and ridge crossing performance as the control target, the mode is switched to low-speed acceleration, hill climbing and ridge crossing, and other operational transition conditions to ensure maximum power and safety performance of the vehicle.

(2)    The impact of anti-skid braking on safety performance

The anti-skid brake is one of the active safety controls designed to avoid excessive wheel slip during vehicle operation. When there is excessive wheel slip, the lateral stability of the whole vehicle will drop sharply, which poses great potential danger to the driving safety of the whole vehicle. Especially after the rain, the ground is slippery, and the different soil water holding capacity, coupled with the unevenness of the ground in the field, leads to agricultural machinery in the driving process in the road conditions. As the road conditions are always changing, for example, the wet soil adhesion coefficient is approximately 0.3 while that of the dry soil is approximately 0.6, it is likely that the wheels on the road surface are not always driving in the same conditions and may thus slip. Adding wheel slip brake control will greatly improve the vehicle's passing and stability performance.

## 6. Conclusions

(1) This paper introduces the structure of the HGPM and the working principle of the power chassis. The mathematical relationship model of the undercarriage transmission system is established. All the parameters of the designed power chassis satisfy the requirements.

(2) Using Solidworks and RecurDyn software, 3D modelling and multi-case simulation of HGPM were performed. According to the theoretical calculation and simulation analysis, the 2H mode can complete a $0-25°$ slope angle climbing and pass over a $0-100$ mm ridge height. The 4H mode can complete a $0-35°$ slope angle climbing and pass through a ridge with height in the range of $0-320$ mm with relatively stable body speed and the wheel rotation angular speed converging faster under the open road condition.

(3) Performance tests were conducted on HGPM under working conditions such as climbing, crossing ridges, and foliation roads. The test shows that under the specific ramp in the $15°$ working condition, the $0-9$ s range is the smooth start to the uniform speed stage, and the $9-20$ s range is the climbing stage. The 4H mode can smoothly pass a ridge with a ridge height of 320 mm and a slope of $26°$, while the 2H mode will have a short fluctuation of speed dropping sharply to 0. The 4H mode effectively improves the steady-state convergence performance of drive efficiency and slip rate compared with the 2H mode, and the overall curves and simulation results are approximately similar.

**Author Contributions:** Z.C. and D.X.: Investigation, Writing an original draft. T.L.: Formal analysis, Data curation. P.H.: Investigation, Supervision. H.L.: Investigation. Q.Z.: Supervision, Funding acquisition. All authors have read and agreed to the published version of the manuscript.

**Funding:** This study was funded by the National Natural Science Foundation of China (Grant No. 52175212).

**Institutional Review Board Statement:** Not applicable.

**Informed Consent Statement:** Not applicable.

**Data Availability Statement:** The data presented in this study are available on request from the corresponding author.

**Conflicts of Interest:** The authors declare no conflict of interest.

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
