# Peer review of "Design of a High-Gap Plant Protection Machine (HGPM) with Stepless Variable Speed and Power Adjustable Function"

_machines, doi:10.3390/machines11020265_

Round 1

Reviewer 1 Report

This paper presents designs an HGPM power chassis transmission system and theoretically analyzes 15 the system based on hydrostatic transmission and time-sharing four-wheel drive (4WD) splitter. However, some comments should be done before publication

1. English structure and typos should be carefully checked and corrected.

2. The main contribution of the paper should be listed in the introduction part of the paper.

3. How do you guarantee the robustness of the power for vehicle?

Author Response

Response to Reviewers

February 2, 2023

On behalf of my co-authors and myself, we would like to thank you for your thoughtful and constructive comments in manuscript number 2143916 entitled "Design Of An High Gap Plant Protection Machine (HGPM) With Stepless Variable Speed And Power Adjustable Function." We believe they have improved this manuscript significantly.

The online resubmission of our manuscript accompanies this letter. We sincerely hope that it can be published in the MDPI Machines.

First and foremost, we would like to thank all the comments and suggestions from the three reviewers and the editor regarding our manuscript, which helped to improve our work. Further, we appreciate the value of our study and the opportunity to resubmit our revised manuscript.

All the issues noted are listed below and addressed individually.

All new changes made to the file are highlighted in red font.

Sincerely Yours,

Zengbin Cai

Issues highlighted by Reviewer 1

This paper presents designs an HGPM power chassis transmission system and theoretically analyzes 15 the system based on hydrostatic transmission and time-sharing four-wheel drive (4WD) splitter. However, some comments should be done before publication

R.: We appreciate your recognition of our work; your encouragement means a lot to us. As the expert said, this paper designed an HGPM power chassis transmission system and theoretically analyzed the system.

Issue 1: "English structure and typos should be carefully checked and corrected."

R.: Thank you very much to the reviewers for their valuable comments, and I apologize for the poor language in our manuscript. We have been working on the manuscript for a long time, and the repeated additions and deletions of sentences and paragraphs have obviously led to poor sentence structure-readability. We have now thoroughly checked the spelling of words and sentence structure in the article and corrected them based on your comments. We sincerely hope that your fluency in the language has improved substantially.

Issue 2: "The main contribution of the paper should be listed in the introduction part of the paper."

R.: Thank you very much for the valuable comments from the reviewers. Your suggestion to add a section on the paper's main contributions in the introduction was felt very necessary and added in the article's introduction section. Modified sections are highlighted in red, as shown in lines 85-96 in the text.

Issue 3: "How do you guarantee the robustness of the power for vehicle?"

R.: Many thanks to the reviewers for their valuable comments. Regarding the issue of vehicle power stability raised by the reviewer, the author chose the EV80-2013037 diesel engine as the power source for the prototype design. The power transmission route is as follows: the engine transmits the power to the HST hydrostatic transmission through the pulley, then transmits the power to the input shaft of the distributor through the gear mesh. Through the distributor, the power is transmitted from the front, and rear drive shaft to the front and rear axle differential, and finally, the power is transmitted to the front and rear wheels by the drive axle to realize the self-propelled operation of the high-ground clearance plant protection machine. The whole process of power transmission is adopted the traditional mechanical structure, the maximum torque of the engine output is 14.3N·m, and the speed is 2100rpm. The maximum input speed of the hydrostatic transmission is 3200rpm, and the splitter and front and rear axle differentials are gear driven. All components meet the normal operating conditions, and the power transmission of the prototype is smooth. In addition, the mathematical derivation of the power transmission process in the power transmission system of the HGPM chassis has been given in the paper. As shown in lines 157-268 of the article.

Reviewer 2 Report

There are some problems in the paper, which should be improved before it is accepted:

The drawbacks of the published research in the field should be presented in more details. The paper contributions should be highlighted versus the previous results, including the authors’ own results.

The authors should use the international abbreviations for units: MPa instead of Mpa, rpm instead of r/min, kW instead of kw etc.

Passive forms are better in academic papers than the active use of "we”.

There are many English language problems that should be corrected. The phrases are long and difficult to follow. A native English speaker should revise the paper. For a good readability, the text should be reorganized in sub-chapters.

Author Response

Response to Reviewers

February 2, 2023

On behalf of my co-authors and myself, we would like to thank you for your thoughtful and constructive comments in manuscript number 2143916 entitled "Design Of An High Gap Plant Protection Machine (HGPM) With Stepless Variable Speed And Power Adjustable Function." We believe they have improved this manuscript significantly.

The online resubmission of our manuscript accompanies this letter. We sincerely hope that it can be published in the MDPI Machines.

First and foremost, we would like to thank all the comments and suggestions from the three reviewers and the editor regarding our manuscript, which helped to improve our work. Further, we appreciate the value of our study and the opportunity to resubmit our revised manuscript.

All the issues noted are listed below and addressed individually.

All new changes made to the file are highlighted in red font.

Sincerely Yours,

Zengbin Cai

Issues highlighted by Reviewer 2

Issue 1: "There are some problems in the paper, which should be improved before it is accepted:  The drawbacks of the published research in the field should be presented in more details. The paper contributions should be highlighted versus the previous results, including the authors' own results."

R.: Many thanks to the reviewers for their valuable comments. The authors have re-summarized and condensed the shortcomings of published studies, paper contributions and results in this area. Revisions have been made in the text and are highlighted in red, as shown in lines 74 -95.

Issue 2: "The authors should use the international abbreviations for units: MPa instead of Mpa, rpm instead of r/min, kW instead of kw etc."

R.: I appreciate the valuable suggestions from the reviewers. I apologize for the substandard unit abbreviations in our manuscript. We have made careful checks and corrections, which have been highlighted in red, as shown in lines 156-159 and 180-181 of the text.

Issue 3: "Passive forms are better in academic papers than the active use of "we"."

R.: We are grateful to the reviewers for their valuable comments. The authors did not consider that the passive form is more relevant in academic papers, so we have checked and revised the whole text, and the revised parts are marked in red.

Issue 4: "There are many English language problems that should be corrected. The phrases are long and difficult to follow. A native English speaker should revise the paper. For a good readability, the text should be reorganized in sub-chapters."

R.: I appreciate the valuable comments from the reviewers, and I apologize for the poor language in our manuscript. We have been working on the manuscript for a long time, and the repeated addition and deletion of sentences and paragraphs obviously led to poor readability. In addition, based on your suggestions, we have extensively revised the syntax of the text and used more short sentences to make it easier to understand. We are now working on both language and readability and have corrected grammatical problems. We really hope that the language will be substantially improved. In response to the reviewer's comment that the article should be reorganized into chapters, the authors have written the manuscript considering that the article is organized to be consistent with the prototype trial process. First, the mathematical theory is derived, and then the simulation is verified. Finally, the prototype is tested, and the sub-chapters of the article correspond to it. After repeated deliberations, the authors concluded that it is more appropriate to align the sequence of the article with the prototype test and test process.

Reviewer 3 Report

I think this manuscript is an interesting topic, and it will be possible to publish it in Machines after the following revisions:

1. The background needs to be improved. At the moment, it is very difficult to assess the reasons for the creation of the presented machine. You do not clearly state the problems with the machines you listed (The phrase "small land size" does not give an understanding of the inapplicability of other machines). It is necessary to clearly formulate the goals and objectives that you set for yourself.

2. I did not find in the manuscript what served as the basis for your machine. Where did the design and layout of the main elements come from? Since you describe one of the problems associated with frequent work on an inclined plane, the layout and arrangement of nodes will have a large impact on the performance of the machine.

3. The text of the manuscript does not contain the technical characteristics of the machine. It is also not clear how it was modeled in SolidWorks.

4. The manuscript does not evaluate the effectiveness of the decisions made in terms of economic parameters.

5. Question on the "Climbing and ridge crossing performance test": You write here about an angle of 15 degrees, but an angle of 25 degrees already appears in the conclusions, where does it come from?

6. Conclusions should be formulated more clearly and everything should be supported by results.

Author Response

Response to Reviewers

February 2 2023

On behalf of my co-authors and myself, we would like to thank you for your thoughtful and constructive comments in manuscript number 2143916 entitled "Design Of An High Gap Plant Protection Machine (HGPM) With Stepless Variable Speed And Power Adjustable Function". We believe they have improved this manuscript significantly.

The online resubmission of our manuscript accompanies this letter. We sincerely hope that it can be published in the MDPI Machines.

First and foremost, we would like to thank all the comments and suggestions from the three reviewers and the editor regarding our manuscript, which helped to improve our work. Further, we appreciate the value of our study and the opportunity to resubmit our revised manuscript.

All the issues noted are listed below and addressed individually.

All new changes made to the file are highlighted in red font.

Sincerely Yours,

Zengbin Cai

Issues highlighted by Reviewer 3

Issue 1: "I think this manuscript is an interesting topic, and it will be possible to publish it in Machines after the following revisions: the background needs to be improved. At the moment, it is very difficult to assess the reasons for the creation of the presented machine. You do not clearly state the problems with the machines you listed (The phrase "small land size" does not give an understanding of the inapplicability of other machines). It is necessary to clearly formulate the goals and objectives that you set for yourself."

R.: We appreciate your recognition of our work; your encouragement is significant to us. We have carefully analyzed the problem of the unclear background elaboration you raised, so we have condensed and rewritten the background of the article to highlight the necessity of prototype creation. The changes have been highlighted in red in the text, as shown in lines 74-83.

Issue 2: "I did not find in the manuscript what served as the basis for your machine. Where did the design and layout of the main elements come from? Since you describe one of the problems associated with frequent work on an inclined plane, the layout and arrangement of nodes will have a large impact on the performance of the machine."

R.:非常感谢您的宝贵意见,本文主要元素和布局参考传统的高地隙植保机,由于HST替换传统的机械变速器和分时四驱分动器替换全时分动器,造成各元器件的布局改变,因此为了平衡整机的质心始终保持在整机的几何中心点,于是将驾驶室布局在机身前端,此外作者增加了计算质心位置的数学理论依据,供审稿专家审核。

主要元素的设计和布局从何而来?因为你描述的是在斜面上频繁工作的相关问题之一,节点的布局和安排将对机器的性能产生很大影响"。

Issue 3: "The text of the manuscript does not contain the technical characteristics of the machine. It is also not clear how it was modeled in SolidWorks."

R.: We appreciate the valuable comments from the reviewers. The article has been supplemented with technical characteristics about the machine and modelling parameters in SolidWorks. The revised parts have been marked in red, as shown in lines 271-274 in the text.

Issue 4: "The manuscript does not evaluate the effectiveness of the decisions made in terms of economic parameters."

R.: We are grateful to the reviewers for their valuable comments. This manuscript addresses the problem of poor adaptability of existing plant protection machines to complex working conditions in the field. An HGPM power chassis drive system was designed, mainly with passability as the evaluation index. Therefore, the manuscript does not really consider the issue of economy, but we will continue to work on the improvement of the upland gap planters with economy as the index in the following work.

Issue 5: "Question on the "Climbing and ridge crossing performance test": You write here about an angle of 15 degrees, but an angle of 25 degrees already appears in the conclusions, where does it come from?"

R.: The authors would like to thank the reviewers for their valuable comments. In the "Climbing and Ridge Crossing Performance Test", the author only had a fixed ramp of 15° for the test due to the limitations of the test environment. The 25° in the conclusion section is explained in line 311 of the paper. However, in the ridge crossing test, the ridge slope was 26°, and the height was 320mm, and the prototype could pass the ridge without any problem, which verified that the prototype could pass the 25° ridge.

Issue 6: "Conclusions should be formulated more clearly and everything should be supported by results."

R.: The authors have rewritten and refined the conclusion section of the manuscript and added quantitative analysis to highlight the accuracy of the conclusion presentation. The changes are highlighted in red in the text, as shown in lines 605 – 622.
